# Polyoxymethylene as Material for Removable Partial Dentures—A Literature Review and Illustrating Case Report

**DOI:** 10.3390/jcm10071458

**Published:** 2021-04-02

**Authors:** Oliver Schierz, Leonie Schmohl, Sebastian Hahnel, Angelika Rauch

**Affiliations:** Department of Prosthodontics and Materials Science, University of Leipzig, Liebigstr. 12, 04103 Leipzig, Germany; leonie.schmohl@medizin.uni-leipzig.de (L.S.); sebastian.hahnel@medizin.uni-leipzig.de (S.H.); angelika.rauch@medizin.uni-leipzig.de (A.R.)

**Keywords:** tooth wear, bruxism, dental restoration wear, resin, synthetic, implant-supported removable partial dentures

## Abstract

Compared to thermoplastic manufacturing techniques, computer-aided design (CAD) and computer-aided manufacturing (CAM) technologies make it easier to process modern restorative and prosthetic materials with improved material properties. In dentistry, tooth-colored alternatives to metal-based frameworks for application in removable dental prostheses (RDP) emerged. With regard to this aspect, the current article provides an overview of the specific material properties of polyoxymethylene (POM). Furthermore, it reviews scientific literature indexed in PubMed and Web of Science that focuses on RDPs fabricated from POM within the last 10 years. Finally, a prosthetic rehabilitation of a patient with a RDP fabricated from POM is illustrated and observations during a follow-up over 10 months are described. Scientific data and clinical observations indicate that polyoxymethylene is a promising material that bridges gaps in dental therapeutic options. While survival time may be limited due to wear, POM might be a favorable option for application in semi-permanent restorations.

## 1. Introduction

Patients demand metal-free and tooth-colored dental restorations for multiple reasons, including sensitivity to alloy components, fear of intoxication, or esthetical reasons. In fixed dental prostheses (FDP), zirconium dioxide is a potent alternative to classic veneered alloys. However, in removable dental prostheses (RDP), metal frameworks are still the gold standard. Polyoxymethylene (POM) is a material able to provide tooth-colored esthetics and that is suitable for fabrication of frameworks for RDPs. POM can be characterized as both polyethers (-C-O-) as well as polyacetals (-O-C-O-) [1]. Two formulations of POM are available: the polyoxymethylen homopolymer (POM-H) and copolymer (POM-C). POM-H can be fabricated either from formaldehyde monomers or from trioxane monomers [1]. POM-H is a highly crystalline thermoplastic material with a helical structure [1,2]. For the formulation of POM-C, small amounts of other cyclic ethers (-C-O-) featuring additional methylene groups are added (Figure 1) [3,4]. The additional methylene groups result in higher thermal and hydrolytic stability, which improves its resistancy against polymer chain degradation [5].

Due to its excellent properties, POM is industrially used as a constructional material, e.g., for the manufacturing of gear wheels, housing parts, and bearings [2,5]. The favorable mechanical properties include high strength, stiffness, hardness, impact strength, low coefficient of friction, high wear resistance, and dimensional stability [2,5,6]. In addition, POM is featured by high chemical resistance, low water absorption, und high biocompatibility [7,8]. The melting point of POM ranges around 175 °C. For thermoplastic manufacturing techniques, injection moulding is the most frequently employed process [8]. By using subtractive manufacturing, artefacts due to smearing can arise due to excessive cutting temperature levels [9]. Additive manufacturing methods such as powder bed fusion have rarely been investigated [6]. Recent literature revealed that POM can also be processed via selective laser sintering [7]. Regarding the optical appearance, POM is characterized by even surfaces and an intrinsic whiteness. The latter is based on its crystallinity [6,7]; however, POM can also be colored [10]. All in all, favorable properties of POM are advantageous for medical application [3,7] and the CAD/CAM-techniques pave the way for application in dentistry (overview see Table 1).

## 2. Literature Review

The popularity of CAD/CAM technologies for the fabrication of RDPs is continuously increasing and traditional manufacturing techniques are increasingly replaced [11]. During the last years, blanks fabricated from polyoxymethylene approved for application as dental restorative materials have become available. Therefore, POM is an interesting material option for application in distinct dental scenarios.

### 2.1. Search Strategy

In scientific dental literature, only little information regarding POM is currently available. Electronic MEDLINE/PubMed and Web of Science searches were performed by using the terms “polyoxymethylene” AND “dentistry” including results from 2010 until 27 January 2021. The exclusion criteria for the search were defined as use in tooth-supported or implant-supported FDP. Moreover, publications focusing on the adhesive bond strength of POM to tooth tissues were excluded (Figure 2). In total, 15 publications were identified by the literature search, which was amended by an additional manual search that retrieved one more reference [12].

### 2.2. Results of Literature Search

The knowledge among dentists about the availability of POM as dental material is sparse [13]. POM shows a hardness reduction of approximately 15 percent after water-storage over 3 months [14]. While staining media such as coffee may cause discoloration of POM, cleansers are effective in removing residues [15]. The flexural strength of POM is lower than that of metal-based alloys fabricated from non-precious metals. Therefore, when using POM as a clasp material, the undercut of the clasp-retaining tooth has to be more than threefold higher than for conventional metal alloys to achieve similar retention forces [16]. Thus, clasps fabricated from POM are often more robustly designed than their metal counterpart. However, if adequately designed, clasps made from POM can be sufficient for clinical use [12]. Regarding reparability, there are shortcomings in comparison to polymethyl methacrylate (PMMA)-based materials as well as metal-based frameworks, as the main problem is the limited chemical interactions with bonding agents [17].

## 3. Case Report

In spring 2019, a 69-year-old male patient introduced himself at the Department of Prosthodontics and Materials Science at Leipzig University (Leipzig, Germany). He described functional impairment related to the orofacial system. However, he was pleased with esthetics and function of the denture in the upper jaw fabricated three years ago. The patient asked for a metal-free prosthetic rehabilitation in the mandible and a minimally invasive treatment approach. Regarding general health, he reported about neurodermatitis, chronic obstructive pulmonale disease, and the insertion of a stent implant in the coronary arteries in 2017.

Clinical inspection of the oral cavity revealed a veneered FDP with a zirconium dioxide based framework in the upper jaw that was characterized by extensive chipping (Figure 3). The upper molars showed signs of moderate periodontitis (Figure 4). In the lower jaw, a severely worn dentition along with reduced vertical dimension of occlusion and multiple carious lesions were obvious. The lower premolars and the right lower second molar had to be extracted. After discussion of various therapeutic options, the patient agreed to extraction of the teeth and a reconstruction of the vertical dimension. Therefore, the latter should be tested with a stabilization splint. For improving retention of a future prosthetic restoration in the lower jaw, the insertion of two implants in the region of the first molars was recommended.

After extraction of the teeth (May 2019), a splint replacing the missing lower teeth was fabricated from translucent methyl methacrylate-free resin (CLEARsplint^®^, Astron Dental, Lake Zurich, IL, USA) (Figure 5).

In September 2019, two implants (Screw-Line Promote Plus 4.3 mm, Camlog, Wimsheim, Germany) were inserted in the area of the lower first molars. At the end of the pre-treatments, following three options for final prosthetic rehabilitation of the lower jaw were discussed with the patient in detail:Removable multi-unit bridge including all remaining teeth and implants, fabricated from zirconium dioxide telescopic crowns and a veneered polyaryletherketone (PAEK) framework,Single-crown fixed dental prostheses for all teeth; Locator abutments on both implants and insertion of a clasp-retained RDP with PAEK framework,Locator abutments on both implants, single-crown FDPs on the remaining lower molar teeth, insertion of a tooth-colored and tooth-shaped monolithic RDP fabricated from POM with minimally invasive preparation of the lower anterior teeth (experimental character in long-term endurance).

The patient refused to be supplied with a new restoration for the remaining lower molar, but preferred the insertion of a tooth-colored RDP.

In January 2020, the sharp edges of the lower anterior teeth were minimally invasively rounded and an impression of both jaws was taken (Figure 6). After the centric relationship was recorded, the RDP was designed (Ceramill Mind v2.4, Amann Girrbach, Pforzheim, Germany) and milled (Ceramill Motion 2, Amann Girrbach, Pforzheim, Germany) from a polyoxymethylene blank (ZirluxAcetal, Henry Schein, Melville, NY, USA). The artificial gingiva was colored and the surface was glazed with light-curing, methacrylate-based materials (OPTIGLAZE^®^ color & clear, GC Europe, Loewen, Belgium) (Figure 7).

After insertion of the locator abutments, the occlusion of the RDP was evaluated. The surfaces of the housings of the locator abutments were sandblasted and intraorally fixed with self-curing luting materials (Quick-up, VOCO, Cuxhaven, Germany). The patient was satisfied with bite comfort and esthetics and received instructions for oral and denture hygiene measurements (Figure 8).

In January 2021, the restoration was reevaluated after 10 months of clinical service. The patient was very pleased with the functional and esthetical rehabilitation. The surface of the glazed material showed no relevant signs of discoloration or increased plaque adhesion. Cleaning and use of the RDP did not produce any major deterioration of the glazing in non-load-bearing areas. Nonetheless, the reevaluation of the occlusal areas revealed relevant signs of wear. The self-curing resin around the housings showed visible discolorations (Figure 9). An overview of the clinical workflow is given in Figure 10, mostly similar to an adjusted splint, but made from a tooth-like colored and shaped material.

In contrast to polyoxymethylene as the main component of the current RDP, most of the other materials that had been used for the fabrication of the denture may produce allergic reactions. Dentists should be aware that coloring and glazing materials contain methylmethacrylate and self-curing luting materials for fixation of the housings contain benzoylperoxide and butylhydroxytoluol (Table 2). However, these materials were not contraindicated in the presented case.

## 4. Conclusions

For application in dentistry, polyoxymethylene does not only feature the advantage of hypo-allergic properties. Even if POM cannot replace traditional materials for RDPs, it is an interesting option for patients with special demands for materials. In this case it was a patient refusing comprehensive preparation of his teeth and who was skeptical regarding alloys. Besides, POM may be the temporary restoration of a worn dentition with tooth-colored material before final restoration. The material is approved for application in RDPs, but limited to provisional crowns and bridges for application in FDPs. As illustrated in the current case report, in patients with severe bruxism, wear might require premature replacement of the restoration. However, POM seems to be an interesting monolithic tooth-colored treatment option for restorations covering all teeth. The material features sufficient color stability and low adhesion of plaque in clinical use. Nonetheless, clinical trials addressing the application of the material in dentistry need to be conducted. The virtual construction and machining of RPDs made from POM using dental CAD-CAM soft- and hardware is a practicable option.

## Figures and Tables

**Figure 1 jcm-10-01458-f001:**
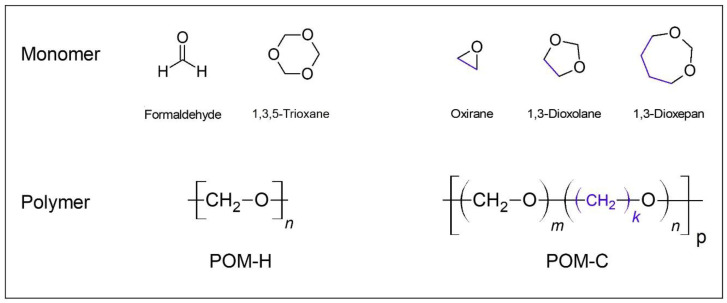
Structural formulas of POM-H, POM-C, and corresponding monomers, additional methylene groups (blue).

**Figure 2 jcm-10-01458-f002:**
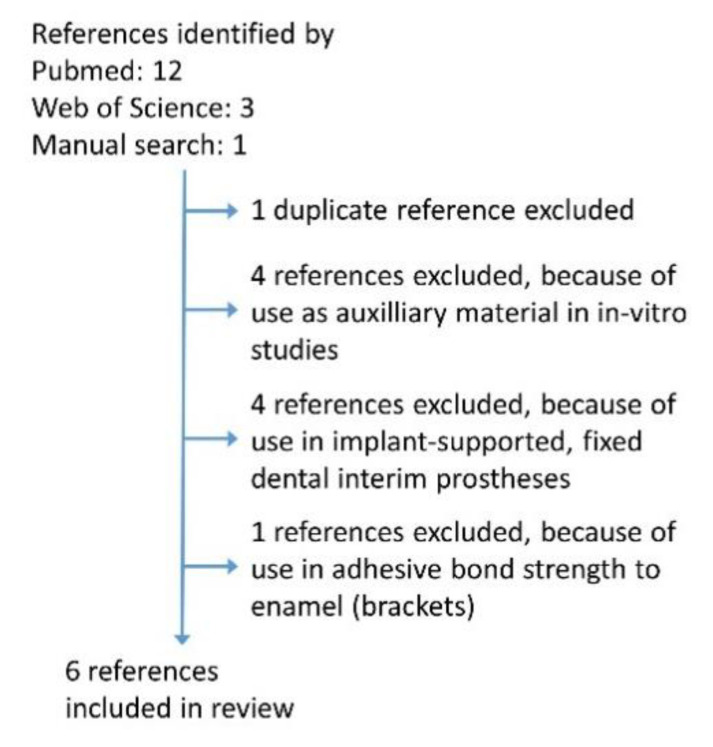
Search strategy.

**Figure 3 jcm-10-01458-f003:**
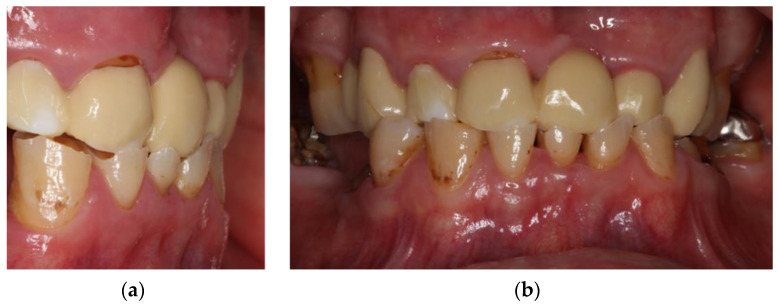
Dental examination in March 2019 revealing a reduced vertical dimension of occlusion, chipping, and severe attrition of the lower anterior teeth: (**a**) lateral view, (**b**) frontal view.

**Figure 4 jcm-10-01458-f004:**
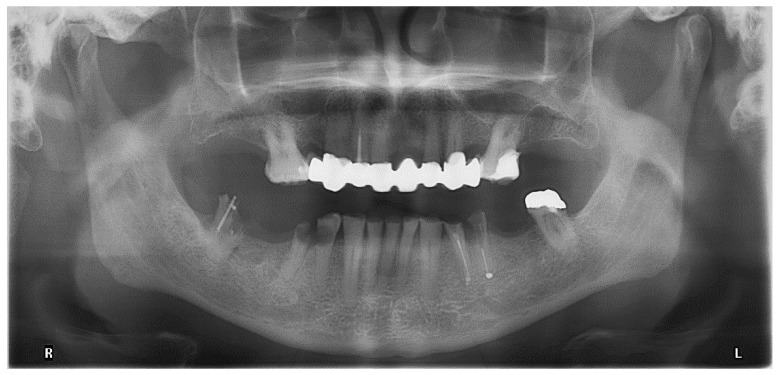
Orthopantomogram presenting the initial oral situation (March 2019).

**Figure 5 jcm-10-01458-f005:**
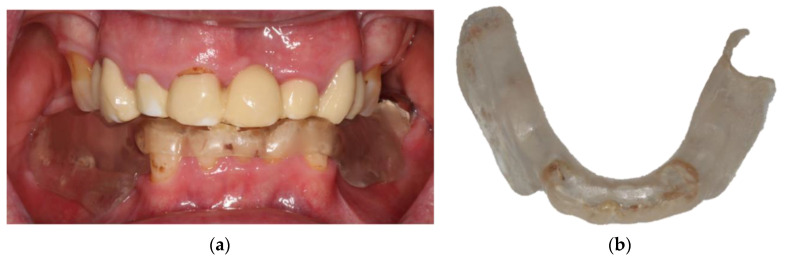
Splint made from translucent resin after 7 months in use: (**a**) intraoral, (**b**) extraoral.

**Figure 6 jcm-10-01458-f006:**
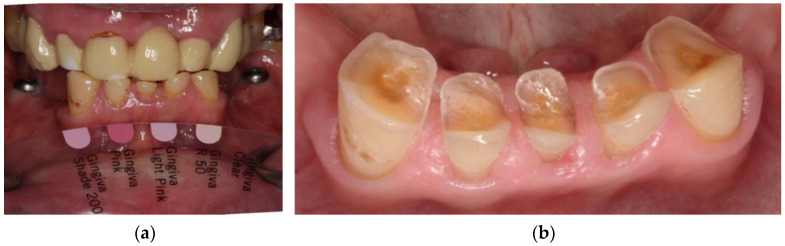
Intraoral situation prior to impression taking: determination of gingiva color (**a**) and preparation design of the lower anterior teeth with rounded edges (**b**). X-rays of the implants placed are available in the Appendix A.

**Figure 7 jcm-10-01458-f007:**
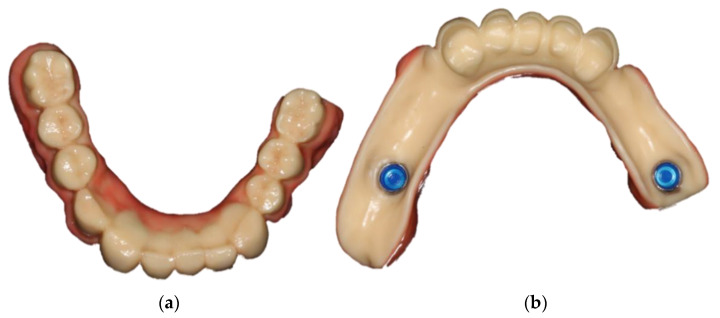
Colored and glazed RDP fabricated from polyoxymethylene: occlusal view (**a**), basal view prior to intraoral fixation of the housings (**b**).

**Figure 8 jcm-10-01458-f008:**
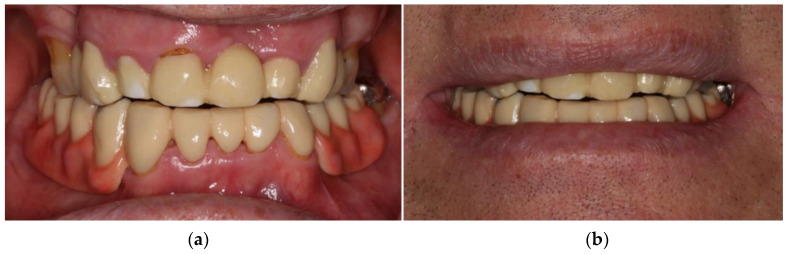
Colored and glazed RDP fabricated from polyoxymethylene shortly after insertion: intraoral view (**a**), extraoral view (**b**). An occlusal view of the lower jaw without denture is available in the Appendix A.

**Figure 9 jcm-10-01458-f009:**
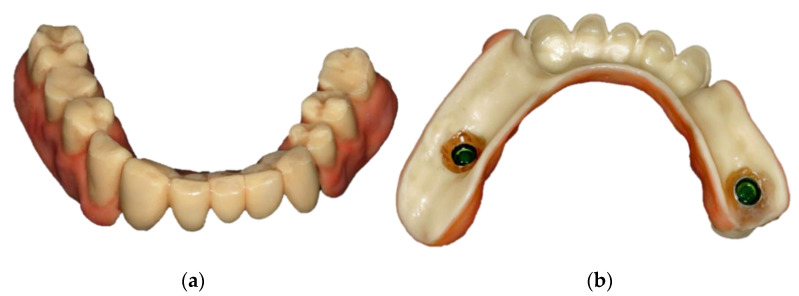
Denture fabricated from polyoxymethylene after 10 months of clinical service: occlusal view (**a**), basal view (**b**).

**Figure 10 jcm-10-01458-f010:**
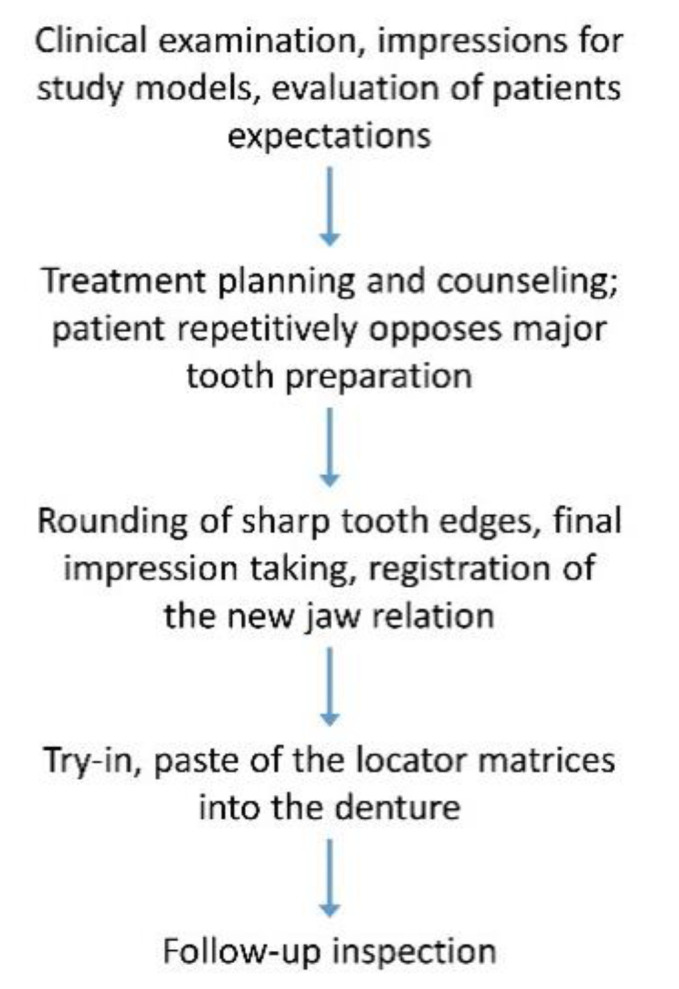
Flowchart of the clinical steps undertaken in the presented case.

**Table 1 jcm-10-01458-t001:** Advantages and Disadvantages of polyoxymethylene as material for removable dentures.

Advantages	Disadvantages
Tooth-colored, available in different shades	Opaque
Break-proof due to high impact strength	Flexible
Smooth surface	Low chemical interaction with other materials
Non known allergies	Limited wear resistance
Color stability	
Customizable color	

**Table 2 jcm-10-01458-t002:** Overview of potentially allergic materials used for fabrication of the denture.

Material	Potential Allergens
Implant, Abutment, Housing	Ti6AI4V ELI, TiN
Matrix insert	Nylon
Light-curing coloring and glazing material	Methacrylate
Self-curing luting material	Hydroxymethacrylate, benzoylperoxide, butylhydroxytoluol

## Data Availability

Data sharing not applicable.

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
