# Peer review of "Polyoxymethylene as Material for Removable Partial Dentures—A Literature Review and Illustrating Case Report"

_jcm, 2021, doi:10.3390/jcm10071458_

Round 1

Reviewer 1 Report

Removable partial dentures made from POM has its limitations in comparison to other materials as referred by the authors, however for  semi-permanent restorations it maybe an interesting material. The authors should  however explain what they understand to be medium-term use in terms of longevity.

Author Response

Thank you for your kind review and suggestion. The material is not limited in intraoral use in general, however in dependence from strain the material will probably show more rapid wear then PMMA-based resins. Therefore, in dependence of the occlusal load a reduced endurance can be estimated. We rewrote the sentence in the conclusion section into “As illustrated in the current case report, in patients with severe bruxism, wear might require premature replacement of the restoration. However, POM seems to be an interesting monolithic tooth-colored treatment option for restorations covering all teeth.”

Reviewer 2 Report

This article is essentially a case report with an analysis of the literature. The clinical case is well described and illustrated and opens perspectives in oral rehabilitation especially for patients with specific needs.

It would be interesting to complete the description of the manufacture of RPD (line 142 page 5): How was the prosthesis designed and milled at the time of manufacture?

On the other hand, the CAD-CAM procedures are mentioned in the introduction but I would also have opened the subject in conclusion. The machining of an RPD with POM is possible with CAD-CAM software such as 3shape or dental wings.

Author Response

Thank you for your kind review and suggestions. We included the requested information into the case report section. “After the centric relationship was recorded, the RDP was designed (Ceramill Mind v2.4, Amann Girrbach, Pforzheim, Germany) and milled (Ceramill Motion 2, Amann Girrbach) from a polyoxymethylene blank (ZirluxAcetal, Henry Schein, Melville, U.S.).”

We now mention the CAD-CAM option in the conclusions. “The virtual construction and machining of RPDs made from POM using general dental CAD-CAM soft- and hardware is now a practicable option.”

Reviewer 3 Report

The manuscript concerns a case report on the use of Polyoxymethylene as material for removable partial dentures. A systematic review has not been performed, (some features are missing to make it so)
The case report presented may be of interest in the prosthetic field as a valid alternative to the present resins and metal ceramics. But there are some considerations that need to be addressed before publishing the manuscript.
Suggestions for authors

• The work is not a systematic review of the literature but rather a literature review. Editing the title removes systematic review and changing it to literature review.
• To make it systematic, a protocol should be followed (for example the PRISMA).
• I would add a summary table of the data in the literature on the advantages and disadvantages of using Polyoxymethylene as material for removable partial dentures
• Is it possible to have a photograph of the intraoral state after the insertion of the implants and immediately before the application of the (RDP)? (Supplements materials)
• Is it possible to have a radiographic image of how the implants were positioned possibly after osseointegration? (supplements materials)
• Is it possible to add a flowchart that summarizes and specifies all the steps and clinical decisions performed in the treatment of the case report?

Author Response

Thank you very much for your thorough review. We changed the title to “literature review”.

We added table 1 containing information regarding advantages and disadvantages.

We added a radiographic image with impression posts placed as well as a photograph of the intraoral state after placement of the locator abutments as supplemental material (figures S1 and S2).

We added a flow chart summarizing the clinical steps (figure 10). The clinical workflow is very similar to the making of an adjusted splint, but made from a different material and shaped like teeth.

Round 2

Reviewer 3 Report

The authors made all the required changes and added the required images in the supplementary materials. I consider the manuscript suitable for publication.

Best regards